# Oral Health Preventive Program in Patients with Autism Spectrum Disorder

**DOI:** 10.3390/children9040535

**Published:** 2022-04-10

**Authors:** Elisabetta Carli, Marco Pasini, Francesca Pardossi, Isabella Capotosti, Antonio Narzisi, Lisa Lardani

**Affiliations:** 1Department of Surgical, Medical and Molecular Pathology and Critical Care Medicine, University of Pisa, I-56126 Pisa, Italy; elisabettacarli1@gmail.com (E.C.); dr.marcopasini@yahoo.it (M.P.); f.pardossi@yahoo.it (F.P.); isacapo@hotmail.it (I.C.); 2IRCCS Stella Maris Foundation, I-56128 Pisa, Italy; antonio.narzisi@fsm.unipi.it

**Keywords:** autism, prevention, oral health

## Abstract

The aim of the study was to evaluate clinical hygienic parameters, patient collaboration, and dental habits in patients with ASD (autism spectrum disorder) before and after a tailored prevention program. A total of 100 patients (78 males and 22 females, mean age 8 ± 0.7 years old) was recruited, with ages ranging from 7 to 16 years old, and diagnoses of ASD. We evaluated the plaque index (IP), gingival index (IG), the dmft/DMFT, the frequency of tooth brushing, and the frequency of snacks for each patient. Patient behaviour was evaluated with the Frankl scale, and each patient was individually reassessed after five visits from the first one by the same operator. The *t* test was used to compare the parameters before and after the inclusion in the dedicated dental pathway. From T1 to T2 we found a significant improvement of the IP (*p* < 0.001), IG (*p* < 0.001), and the frequency of tooth brushing (*p* < 0.001). Concerning the frequency of snacks and the parameter dmft/DMFT, the differences in the observed averages were not significant (*p* > 0.05). The difference in collaboration between T1 and T2 evaluated by the Frankl scale was statistically significant (*p* < 0.001). It was found that the prevention program allowed a significant improvement in both clinical parameters and patient behaviour. The personalized digital supports can have a key role for success in familiarization and desensitization processes of patients affected by ASD, leading an increase in their collaboration.

## 1. Introduction

Autism spectrum disorder (ASD) is a severe multifactorial disorder characterized by an umbrella of specific symptoms in the areas of social communication, restricted interests, and repetitive behaviours. ASD varies greatly in the severity of associated socio-communicative impairments and in the degree of cognitive and language development [1].

This disorder has a multifactorial aetiology and onset in the first three years of life; Kanner introduced the term autism in 1943, and, concerning the nosographic classification level, in the DSM-V (*Diagnostic and Statistical Manual of Mental Disorders*, 5th ed.), ASD belongs to the clinical category of neurodevelopmental disorders [2].

The prevalence of ASD is worldwide, and recent epidemiological data estimated it to be 1/44 in United States and 1/87 in Italy [3,4].

Even if autism does not have a direct effect on oral cavity health, the failure to correctly apply oral hygiene rules and the difficulties of these patients to undergo dental visits and therapies exposes them to greater risk of development of carious lesions, alteration of the periodontal status, alteration of the oral microbiota, and an increased risk of traumatic injury [5,6]. Through a systematic review and meta-analysis, the prevalence of dental caries was found to reach 60.6%, while the prevalence of periodontal disease was 69.4% [7]. It was observed that children with ASD often behaved negatively, or definitely more negatively than healthy children without ASD during dental visits; therefore, therapeutic compliance can be challenging to achieve. Furthermore, it was found that self-injurious behaviour and bruxism were frequent in children with ASD, and their behaviour and other life factors may complicate provision of dental services and limit access to paediatric dental care [8].

Unfortunately, there is a tendency by the relatives, educators, and operators to focus the attention almost exclusively on the neuropsychiatric aspects of the patient’s pathology, minimizing the possible damages affecting the oral cavity. These problems are being dealt with only in moments of urgency, when it is too late to obtain the desired affects, resulting in the halo effect. 

Since ASD is a heterogeneous disease with a wide range of expressions in individuals, appropriate and specific strategies are needed. The dental management of the person with ASD requires a gradual multi-step approach to scrupulously plan all actions to be carried out before, during, and after the outpatient treatment to obtain a personalized program aimed at reducing or completely removing the mental, spatial, and perceptual obstacles of each. Through forms of verbal and non-verbal communication, gradual contact has the goal to progressively familiarize the patient with the tools and the environment of care. This so called “inactive” phase is followed by the “active” phase, consisting of the “Tell, Show, Do” module [9]. This module allows the best use of the subject’s most elementary and primary intellectual resources, favouring a positive association between the dental clinic’s environment and the absence of danger and pain [10]. Communication is the passage of information between a person and the world, which necessarily consists of different subjects who can interact through this extraordinary means. If the person with ASD does not have language, communication must be sought on the acoustic, visual, and tactile levels [11]. For this reason, technologies make it possible to use a purely visual channel for communication that is not only clear, structured, and predictable but that also does not cause emotional inferences or incomprehensible implications for a person with ASD. The usefulness of this approach has been demonstrated and validated, making information technologies valid supports in intervention and treatment programs with the final goal of enhanced learning, communication, socialization, and patient collaboration [12]. More specifically, during each dental session, multimedia material could be collected and processed to create digital interactive social clips of the meeting’s key point.

The aim of the present study was to evaluate the efficacy of preventive programs with the aid of dental software for the improvement of clinical parameters and patient collaboration in paediatric patients with ASD.

## 2. Materials and Methods

The study was conducted on children and adolescents with ASD with an age ranging from 7 to 16 years who had been diagnosed with autism spectrum disorder by local child psychiatry units. The standardized protocol included the evaluation of autism severity throughout ADOS-2(Autism Diagnostic Observation Schedule-Second Edition), and language development and cognitive and adaptive functioning. All subjects with ASD were native Italian speakers.

All subjects were treated with the aid of application software. The sample was selected based on specific exclusion and inclusion criteria.

The inclusion criteria were as follows: diagnosis of autism spectrum disorder, permanent or mixed dentition, patients on whom it was possible to carry out a re-evaluation after at least five sessions with a Frankl scale score >1, and patients whose parents had signed an informed consent form. The exclusion criteria were as follows: patients on whom it was not possible to carry out five sessions within the course because they were a priori not able to come to the visits, patients with a Frankl scale score <2, patients who had received dental hygiene sessions during the previous 3 months, and patients with systemic diseases.

All procedures were conducted according to the principles expressed in the Declaration of Helsinki and the Italian Association of Psychology (AIP); furthermore, ethics approval was obtained by the local ethics committee (Prot. N. 10–06/05/2019 AOUP, Pisa, Italy).

Before performing the first dental visit, an interview was carried out with the patient’s parents. During the interview, an anamnestic file was drawn up together with the data collection file with the child neuropsychiatrist’s collaboration to better characterize the neuropsychiatric aspect of the patient.

Dental visits were performed at T0 and T1 (after 1 week), and two more visits were performed after one month from T1 and after three months from T1. Moreover, after 6 months from T1, a final visit was performed (T2). Clinical parameters were evaluated only at T1 and T2. Each dental visit had an average time of 30 min to 45 min.

The first visit (T0) could be defined as a “cognitive” visit: a fundamental moment for the patient to know the dental team and the clinic’s settings (roles, methods, and times). During the first visit, it was often possible to familiarize the patient with dental instruments and to record these moments through photos and videos subsequently sent to the family. At the end of T0, some objects, such as a mirror, and a toothbrush, were given to the patient as visit reinforcement and to improve confidence with the tools. Moreover, the first visit improved patient compliance, and all subjects with a Frankl score of 4 decreased their score after 1 week (T1).

After 1 week, a second visit was performed (T1) with the goal for the patient to become more directly familiar with the dental instruments through the “Tell, Show, Do” module. Moreover, other behavioural approach techniques were used, i.e., voice control (alterating voice volume), distraction technique (diverting the patient’s attention from what may be perceived as an unpleasant procedure), and enhancing control (raising a hand in order to interrupt the dental procedure).

During the visit, the functioning of the several dental tools was explained and shown by the dentist by gently testing the instruments, i.e., on the patient’s nail. Through the demonstration of the usage of the probe and the mirror, an objective examination of the oral cavity was carried out, and the following parameters were evaluated at T1 and T2:
Brushing frequency: evaluated with an index range of 1 (which corresponds to several times a week), 2 (once a day), and 3 (twice a day).Frequency of snacks/sugary drinks: measured through an index of 1 (≤1/day), 2 (≤2/day), and 3 (≥3/day).Plaque index (IP): assessed according to Silness and Loe [13] evaluating the presence or absence of four surfaces around each tooth (mesial, distal, buccal, and lingual) and giving a score ranging from 0 (absence of dental plaque) to 3 (severe accumulation of dental plaque). Then, the mean and standard deviation of the total sites were calculated. For a better assessment of the plaque, an erythrosine plaque detector was used.Gingival index (GI): a value from 0 to 3 was assigned for the four sites of each element (mesial, distal, vestibular, and lingual/palatal) in increasing order of inflammation. To determine the GI in each individual, all the values of each tooth divided by the number of teeth examined were added.dmft/DMFT (decayed, missing, and filled teeth): index that targeted the incidence of past and current caries in a population with permanent/deciduous dentition. To calculate the percentage in each child, the decayed teeth, the missing teeth, and the filled teeth were added up, and the sum was divided by the number of teeth.Frankl scale: behavioural scale based on a classification that included 4 levels of collaboration: 1 (definitely negative), 2 (negative), 3 (positive), and 4 (definitely positive).


Regular visits were performed in order to increase patient collaboration after 1 month and after 3 months from T1. During these visits, the functions of rotating instruments were shown to the patient through the “Tell, Show, Do” module by using the gradual approach. In fact, instruments were first shown outside the oral cavity and then tested on a nail, switching them to on and off. Subjects were asked to bring their own toothbrush to the visit for a gradual approach to the correct oral hygiene manoeuvres that were initially performed with a personal toothbrush and then with professional tools. It was possible to carry out prevention and treatment manoeuvres that corresponded to moments in which it was very important to respect every single sequence. In this phase, video modelling and sequencing games with interactive pdf were used to clearly outline the order of execution of the various actions. At the end of the session, the device played the role of positive reinforcement. In fact, a video was shown on the tablet both as a reward for collaboration in the session just ended, and as a positive reinforcement for the following sessions. Another important aspect that the tablet covered was the reinforcement at home. This process was made possible thanks to patients’ parents, who repurposed at home the moments captured during the child’s visits in the clinic, significantly increasing the child’s compliance.

T2 was the evaluation date in which the parameters taken into consideration at T1 were re-evaluated to assess, according to these indexes, the improvement of the patient’s health and collaboration.

### Statistical Analysis

Sample size calculation was based on IP score of a previous study [14].

In order to compare the two means with a power of 0.9, a size of the test of 1%, a standard deviation of 0.8, and a difference of 0.7, the sample size required 23 patients in each group.

Pearson’s Correlation Coefficient was performed to evaluate the correlation between brushing frequency and the number of times the video was shown at home.

The means and standard deviations in the sample of 100 children at T1 and T2 (after 6 months) were calculated with a specific statistical program SPSS 22.0 (SPSS Inc, Chicago, IL, USA). Measurements were compared using the paired *t* test for normally distributed variables and the Wilcoxon test when the assumption of normality was not complied with. The variation between T1 and T2 were measured, and the *p*-value was set at the level of *p* < 0.01.

## 3. Results

This study was conducted on 100 patients (78 males and 22 females, mean age 8.2 ± 2.6 years old) affected by ASD at the Paediatric Dentistry University Clinic, specifically from the clinic that deals with autism spectrum disorder. For each parameter, Table 1 reports the mean and the standard deviation related to the evaluation (T1) and the re-evaluation (T2) dates.

The mean of the brushing frequency significantly improved between T1 and T2 (*p* < 0.0001). The video was shown at home at least three times per week for each subject (mean value: 3.1 ± 0.1), and a strong correlation was observed between the improvements and the number of times the video was shown at home (degree of correlation 0.89; *p* < 0.001).

As regards clinical oral indexes, the plaque index showed a lower value at T2 in comparison to the first visit (*p* < 0.0001); moreover, the gingival index was lowered with an average of 2.03 in T1 up to 1.34 in T2, leading to an improvement in oral hygiene (*p* < 0.0001). However, the frequency of snacks did not improve significantly compared to the first visit (*p*: 0.0287).

Furthermore, the dmft/DMFT value increased between T1 and T2, and no significant difference was recorded (*p*: 0.0644). However, the specific program led to a statistically-significant improvement of the Frankl scale, reaching almost the top degree of the scale, from 2.04 in T1 to 3.2 in T2 (*p* < 0.0001).

## 4. Discussion

Brushing frequency was significantly improved in the re-evaluation session in comparison with the first one. In addition to the medical approach, the role of parents in the protocol was crucial. In fact, they were the most important resource in promoting child behaviour changes, since they performed the training on their child at home after being trained. A significant improvement between the number of times the video was shown and brushing frequency was observed.

As the child with ASD strongly feels family influence, because of its heavy dependence on the parents, good parent motivation and active involvement in the entire therapeutic path brings good results, especially to domiciliary oral hygiene [15]. Parents were involved in the preliminary interview where, besides for an anamnestic folder, an additional folder about specific features of the child was compiled. Knowledge of positive and negative reinforcements, custom practices, and the communication channel is important [16].

The differences in frequency of sugary snack drink intake between T1 and T2 were non-significant and can be explained mainly by the difficulty in changing mealtime routines; furthermore, new healthy foods may cause anxiety in ASD children, and these children have often preferences for unhealthy foods.

As regards the fact that children with ASDs go to daytime centres where educators sometimes resort to edible reinforcements such as chocolate, candy, and chips to stimulate and reward good children’s behaviour, we should highlight that extremely small quantities of snacks are recommended to be used, often in the size of millimetres; furthermore, the amount and frequency of sugar intake during the treatment should be constantly reduced. This practice mostly happens in the initial phase of Applied Behaviour Analysis (ABA) therapy, and even if it may be useful to adopt other reinforcers than sweet reinforcers, this procedure is often very hard to achieve [17,18]. Therefore, the benefits of ABA therapy outweigh other possible disadvantages.

Oral health preventive procedures including dental health nutrition, correct brushing techniques, and fissure sealants should be performed in ASD children to decrease caries incidence [19,20].

The dmft/DMFT results were non-significant. It has been shown in the literature that children with ASD do not have a greater susceptibility to caries or periodontal disease in comparison with the children without ASD, and their worst state of health depends exclusively on the greater difficulty in practicing correct oral hygiene methods and on the limited access to prevention and treatment facilities [21,22].

In fact, in our study only based on preventive programs and education rather than therapeutic procedures, we found that with a decrease in plaque and gingival index and a higher brushing frequency, there was not a significant increase in the prevalence of decayed, missing, or filled teeth (dmft/DMFT index), even if a slight worsening trend was found. Therefore, dental prevention programs are necessary for ASD children even if the development of new caries lesions in these patients is very hard to prevent.

Preventive dentistry should be performed as soon as possible to improve good oral healthcare habits in order to obtain a decrease of enamel and periodontal lesions.

Based on each different patient’s case, oral hygiene manoeuvres were performed in the first sessions. In fact, from T1 to T2, we found a significant improvement of both plaque and gingival indices. The application support was helpful at home because through interactive pdf and video modelling, the child improved both home oral hygiene procedures and the level of collaboration. This represented an essential goal for our study, since it aimed to demonstrate how a dedicated and specific approach increased each ASD patient’s level of compliance.

In this protocol we also evaluated the Frankl scale, a behavioural scale based on a classification that includes four levels of collaboration. The difference between T1 and T2 evaluated by the Frankl scale was statistically significant. This result showed that the dedicated path led to a better degree of compliance. Therefore, sedative use or general anaesthesia may remain necessary only in cases of extreme urgency. Among the limits of the present study, we did not use the ICDAS index, which is a more specific and dynamic index in comparison to dmft. Moreover, we could not detect the real effect of personalized digital supports, as both digital and traditional prevention programs were performed without comparing two different groups. Another limit of the study is the absence of a control group of healthy patients without ASD in order to evaluate if the results found in an ASD population could be similar to that of the general paediatric population. The tested protocol allowed the most common dental treatments to be performed in children with ASDs, which are usually carried out in young subjects, without sedative or general anaesthesia.

## 5. Conclusions

Results showed that it is possible to bring the ASD paediatric population to a better state of oral health with a specific oral health preventive program. In the present study we found a significant improvement of clinical parameters (PI and GI), and the prevalence of decayed, missing, or filled teeth (dmft/DMFT) did not significantly increase at the end of the study. Moreover, tooth brushing frequency was significantly higher at T2, and the frequency of snacks during the day was significantly lower at the final follow-up.

The Frankl scale of behaviour related to dental intervention significantly improved after six months; therefore, it is hoped that the first dental visit of children with ASDs will occur immediately after the disorder’s diagnosis to focus more on prevention and not on therapeutic interventions, which are difficult to perform in young patients with anxiety and fears.

To conclude, there is a need to extend individualized projects throughout the territory to allow children with ASDs to be treated with a specific oral health preventive program.

## Figures and Tables

**Table 1 children-09-00535-t001:** Mean and standard deviation at T1 and T2 (after 6 months).

Parameters	Mean		Standard Deviation		*p*-Value
	T1	T2	T1	T2	
Brushing frequency	2.29	2.88	0.85629	0.35619	0.0001
Frequency of snacks	1.05	0.87	0.67232	0.46395	0.0287
Plaque index (IP)	3.20	1.41	0.76723	0.55433	0.0001
Gingival index (GI)	2.03	1.34	0.80973	0.80679	0.0001
dmft/DMFT	0.11	0.15	0.14373	0.14688	0.0644
Frankl scale	2.04	3.20	0.75103	0.66667	0.0001

## Data Availability

The data supporting the findigs of this study are avaible upon reasonable request.

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
