# Peer review of "Oral Health Preventive Program in Patients with Autism Spectrum Disorder"

_children, 2022, doi:10.3390/children9040535_

Round 1
Reviewer 1 Report
Comment
(1) This study addressed the important topic of dental care for children on the autism spectrum.
(2) the applied research methodology allowed us to find a method for improving oral hygiene care of children from the autism spectrum
(3) The role of parents in the trainings deserves special emphasis, which was made in the reviewed article in the discussion of results.
(4) An important conclusion of the article is the harmfulness of snacks used as rewards within the applied ABA therapy. It is worth mentioning that this leads to other eating disorders.
(5) the reference to the ICD-11 criteria needs supplementation. the article is written in Europe and will be published here. Reference only to DSM-5 seems insufficient.
Author Response
(1) This study addressed the important topic of dental care for children on the autism spectrum.
Thank you for your comment.
(2) the applied research methodology allowed us to find a method for improving oral hygiene care of children from the autism spectrum
Thank you for your comment.
(3) The role of parents in the training deserves special emphasis, which was made in the reviewed article in the discussion of results.
Thank you for your comment.
(4) An important conclusion of the article is the harmfulness of snacks used as rewards within the applied ABA therapy. It is worth mentioning that this leads to other eating disorders.
- it was written that “We highlighted the harmfulness of snacks used as rewards within the applied ABA therapy; this may lead to other eating disorders i.e.: bulimia nervosa”.
(5) the reference to the ICD-11 criteria needs supplementation. the article is written in Europe and will be published here. Reference only to DSM-5 seems insufficient.
- references were modified according to the new sentences. Prevalence of ASD were added and incidence was removed.
Reviewer 2 Report
Dear authors, unfortunately, despite dealing with an interesting topic, this work needs substantial revisions before it can be evaluated for publication.
- some bibliographic entries are missing
• lines 27-39 the historical introduction on autism is not suitable for a pediatric journal in which readers are assumed to have already understood these concepts
• introduction it would be suitable to deepen the behavioral approach techniques and which of these, including visual pedagogy, have been usedin the literature
• introduction it would be suitable to talk about modeling, its evolution, videomodeling and PECS (techniques on which the study design is based)
• lines 84-85 is not clarified that the objective of the study is limited to a population of pediatric autistic patients
• Patient numbers and demographic characteristics are results and therefore would go in the results section and not materials and methods
• the sample included had children between the ages of 7 and 16 among the criteria, how is it possible that the SD of the mean age is so low?
• lines 91-92, which other route is meant?
• lines 94-96 permanent and mixed dentition is repeated among the inclusion criteria
• line 97 "patients on whom it was not possible to carry out five sessions within the course" is not clear if are the patients who did not come to follow up and were excluded or if are patients a priori not able to come to the visits - line 101 ethical approval has been cited but no committee or protocol number has been cited
• lines 115-117 is not clear if were used other behavioral approach techniques
• lines 106-107 is confusing, the timing of the search and appointments is not clarify and can not be reproduced. the time of the study is not reported.
• comparing t2 with t1 in patients whose collaboration assessed with the Frankl scale significantly improved, how was it possible to evaluate 4 sites per dental element of all elements for both PI and GI in children with g4 at T1?is not clarified
• line 134 is not clear why has a caries index such as dmft been evaluated, which cannot decrease but only increase. Increasing the number of times the elements are washed with fluorinated toothpastes it would have been interesting to evaluate the status of carious lesions with more specific and dynamic indices such as the ICDAS
• line 125 there are several PIs described in the literature (Silness & Löe, O’Leary etc ...) which one has been selected for the study is not specified
• lines 162 shows the parameters at T0, while materials and methods mean that the parameters are evaluated at T1
• lines 160-163 the statistical analysis is weak, and a description of the sample at T0 and T2 is missing
• line 171 "The plaque index 171 (IP) improvement at final follow-up was significant (p <0.001)" the parameter shown in the table has increased, is not clear how this is an improvement and not a worsening
• line 173, if in accordance with the Frankl scale higher values are related to a negative collaboration the sentence "The Frankl scale improved on the re-evaluation date with a statistical significant difference (p <0.01)" with an increase in the values a T2 compared to T1 is not clear
• line 179 "plaque index showed a lower value at T2 in comparison to the first visit (p <0.01)" The values in the table are greater than T2
• lines 184-186 is not clear according to the previous definition of collaboration degrees of Frankl scale given above
• lines 189-194 no correlation was made between the improvements and the number of times the video was shown
• lines 207-213, the absence of a statistically significant difference between T1 and T2 is positive as the dmft could only increase, and despite the continuous visits it increased, no explanation is given
• line 220 interactive pdf, has never been mentioned before in the text, the materials and methods section are lacking and incomplete
• the study limits are missing in the discussion and conclusions section
Author Response
First of all, thank you for your comments and suggestions that allowed us to greatly improve the quality of the manuscript.
- some bibliographic entries are missing
• lines 27-39 the historical introduction on autism is not suitable for a pediatric journal in which readers are assumed to have already understood these concepts- historical introduction was modified and some sentences were removed.
- historical introduction was modified and some sentences were removed.
- introduction it would be suitable to deepen the behavioral approach techniques and which of these, including visual pedagogy, have been used in the literature
- sentences including visual pedagogy were written in the new manuscript.
- sentences including visual pedagogy were written in the new manuscript.
- introduction it would be suitable to talk about modeling, its evolution, videomodeling and PECS (techniques on which the study design is based)
- new sentences including modeling, its evolution, videomodeling and PECS were added in the text
- new sentences including modeling, its evolution, videomodeling and PECS were added in the text
- lines 84-85 is not clarified that the objective of the study is limited to a population of pediatric autistic patients
- it was written that the objective of the study was limited to a population of pediatric autistic patients
- it was written that the objective of the study was limited to a population of pediatric autistic patients
- Patient numbers and demographic characteristics are results and therefore would go in the results section and not materials and methods
- Patient numbers and demographic characteristics were moved to results.
- the sample included had children between the ages of 7 and 16 among the criteria, how is it possible that the SD of the mean age is so low?
- mean and SD were not correct and we put the right sentence: mean age 8.2 ± 2.6 years old)
- lines 91-92, which other route is meant?
- he sentence was not correct and a new sentence was written “All subjects were treated with the aid of an application software”.
- lines 94-96 permanent and mixed dentition is repeated among the inclusion criteria
- repetition were removed from the text.
- line 97 "patients on whom it was not possible to carry out five sessions within the course" is not clear if are the patients who did not come to follow up and were excluded or if are patients a priori not able to come to the visits
- it was written that they were not able to come to the visits and patients with a Frankl scale score 1 (definitely negative). Moreover, Frankl scale definition was not correct in the materials and methods and it was modified.
- line 101 ethical approval has been cited but no committee or protocol number has been cited
- protocol number was added in the new manuscript
- lines 115-117 is not clear if were used other behavioral approach techniques
- other behavioral approach techniques were used i.e: voice control, distraction and enhancing control.
- lines 106-107 is confusing, the timing of the search and appointments is not clarify and can not be reproduced. the time of the study is not reported.
- Patients were visited at T0, T1 (after 1 week) and two more visits were performed after one month from T1 and after three months from T1. Moreover, after 6 months from T1 a final visit was performed (T2). Clinical parameters were evaluated only at T1 and T2. The paragraph was added.
- comparing t2 with t1 in patients whose collaboration assessed with the Frankl scale significantly improved, how was it possible to evaluate 4 sites per dental element of all elements for both PI and GI in children with g4 at T1?is not clarified
- it was highlighted in the inclusion and exclusion criteria that patients with a Frankl scale score < 2 were not included in the study.
- line 134 is not clear why has a caries index such as dmft been evaluated, which cannot decrease but only increase. Increasing the number of times the elements are washed with fluorinated toothpastes it would have been interesting to evaluate the status of carious lesions with more specific and dynamic indices such as the ICDAS
- dmft was evaluated because the primary objective of the study was to evaluate the improvement of patients collaboration and dental hygiene indexes. ICDAS is a more specific and dynamic index than dmft but the aim of the study was not to measure the effect of fluorinated toothpastes. Moreover, toothpastes use was not constant among all patients.However, we added the lack of ICDAS index among the limits of the present study.
- line 125 there are several PIs described in the literature (Silness & Löe, O’Leary etc ...) which one has been selected for the study is not specified
- it was written that “it was assessed according to Silness and Loe”
- lines 162 shows the parameters at T0, while materials and methods mean that the parameters are evaluated at T1
- it was written that all parameters were evaluated at T1.
- lines 160-163 the statistical analysis is weak, and a description of the sample at T0 and T2 is missing
- statistical analysis was improved. Sample size calculation was added in the manuscript. Pearson’s Correlation Coefficient was performed to evaluate the correlation between brushing frequency and the number of times the video was shown at home. We wrote that program SPSS 22.0 (SPSS Inc, Chicago, IL, USA) was used. Moreover it was written that “Measurements were compared using the paired t test for normally distributed variables and the Wilcoxson test when the assumption of normality was not complied with. The variation between T1 and T2 was measured ...”. As regards sample description it was moved from materials and methods to the result section “100 patients (78 males and 22 females, mean age 8.2 ± 2.6 years old) affected by ASDs”. Moreover, in the statistical analysis it was stated that: “The means and standard deviations in the sample of 100 children at T1 and T2 (after 6 months) were calculated...”.
- line 171 "The plaque index 171 (IP) improvement at final follow-up was significant (p <0.001)" the parameter shown in the table has increased, is not clear how this is an improvement and not a worsening
- plaque index improved at the final follow up; mean and SD were not correct in the table and they were modified in the new table.
- line 173, if in accordance with the Frankl scale higher values are related to a negative collaboration the sentence "The Frankl scale improved on the re-evaluation date with a statistical significant difference (p <0.01)" with an increase in the values a T2 compared to T1 is not clear
- we have modified Frankl scale definition in the materials and methods section: Frankl scale: behavioural scale based on a classification that included 4 levels of collaboration: 1 (definitely negative), 2 (negative) 3 (positive) and 4 (definitely positive).
- line 179 "plaque index showed a lower value at T2 in comparison to the first visit (p <0.01)" The values in the table are greater than T2
- plaque index improved at the final follow up; mean and SD were not correct in the table and they were modified in the new table.
- lines 184-186 is not clear according to the previous definition of collaboration degrees of Frankl scale given above
- we have modified Frankl scale definition in the materials and methods section: Frankl scale: behavioural scale based on a classification that included 4 levels of collaboration: 1 (definitely negative), 2 (negative) 3 (positive) and 4 (definitely positive).
lines 189-194 no correlation was made between the improvements and the number of times the video was shown- In Discussion it was added: “A significant improvements between the number of times the video was shown and brushing frequency was observed”. Moreover, in the Results it was written that “The video was shown at home at least three times per week for each subject and a strong correlation was observed between the improvements and the number of times the video was shown at home (degree of correlation 0.89)”. In the statistical analysis a specific sentence was added.
- lines 207-213, the absence of a statistically significant difference between T1 and T2 is positive as the dmft could only increase, and despite the continuous visits it increased, no explanation is given
- it was clarified that: “Despite the absence of a statistically significant difference between T1 and T2 was positive, dmft index slightly increased at the final follow-up probably because snacks consumption frequency was still moderate at T2”.
- line 220 interactive pdf, has never been mentioned before in the text, the materials and methods section are lacking and incomplete
- in materials and methods it was added that: “In this phase video modeling and sequencing games with interactive pdf were used-...”
- the study limits are missing in the discussion and conclusions section
- study limits were added in the discussion and conclusions.
Reviewer 3 Report
Dear Authors,
I appreciate very much that you share your good experience. Find attached the comments and suggestions.
Best regards.

Author Response
First of all, thank you for your comments and suggestions that allowed us to greatly improve the quality of the manuscript.
Title should indicate a prevention program related to oral health is the topic.
- the title was modified including “oral health”
Line 9 (Sentence 1) Please change the statement into a regular sentence.
- the sentence was modified into: “The aim of the study was to evaluate clinical hygienic...”
Lines 27-29 Please provide an accurate definition/description of ASD. The core symptoms include impairments of social and communication skills and repetitive behaviors. In fact, the other symptoms are comorbidities, that may be present, but not in all individuals, and they are not a characteristic of ASD.
- a more accurate description of ASD was added in the introduction: Autism spectrum disorder (ASD) is a severe multifactorial disorder characterized by an umbrella of specific symptoms in the areas of social communication, restricted interests, and repetitive behaviors [1].
Lines 34-37: Diagnostic criteria and their development/changes are a complex topic, and the facts are not precisely described. Since they are not directly related to the topic of the paper I recommend to reduce this section. In this context reference (2) is irrelevant. I suggest rather to extend the next section and introduce more broadly the oral health behaviors of children with ASD and barriers to dental care.
- Diagnostic criteria section was reduced while oral health behaviors in children was extended “It was observed that children with ASD behaved negatively or definitely negatively than healthy children during dental visits; therefore, therapeutic compliance could be challenging to achieve. Furthermore, it was found that self-injurious behaviour and bruxism were frequent in children with ASD and their behaviour and other life factors may complicate the provision of dental services and limit access to pediatric dental care [8]”.
- Reference 2 was deleted.
Lines 39 – 40 The resource ISTAT 2018 is not included in the list of references. Therefore according to other resources I can only presume that the authors wanted to indicate that the „... in the past, incidence rates have been increasing by 10% in a year to currently reported increase by 17% per year“. E.g. if previously the yearly change was form 10 per 100 000 to 11 per 100 000, now it would be an increase to 1,17/100 000. This does not mean that the incidence was 10 % and it increased to 17%. Please add the reference into the list.
- Incidence was removed while prevalence was added. References 3 and 4 were added “ 3. Maenner MJ, Shaw KA, Bakian AV, et al. Prevalence and Characteristics of Autism Spectrum Disorder Among Children Aged 8 Years — Autism and Developmental Disabilities Monitoring Network, 11 Sites, United States, 2018. MMWR Surveill Summ 2021;70:1-16. and Narzisi A, Posada M, Barbieri F, Chericoni N, Ciuffolini D, Pinzino M, Romano R et all. Prevalence of Autism Spectrum Disorder in a large Italian catchment area: a school-based population study within the ASDEU project. Epidemiol Psychiatr Sci. 2020;29:1–10”.
Line 73: It is recommended to prefer the term „the person with ASD“ instead of „autistic person“. The latter is sometimes perceived as rude. Instead autistic person does not have language, consider the term „nonverbal person/individual“.
- ASD was used in the new manuscript.
Line 94 Please indicate who diagnosed the patients (psychologist, psychiatrist, etc. – depending on the standards in your country ), also id they were diagnosed by standard methods.
- the following sentence was added in the text: by local Child Psychiatry Units. The standardized protocol included the evaluation of the (a) autism severity throughout ADOS-2; language development; and cognitive and adaptive functioning. All subjects with ASD were native Italian speakers.
Line 120: Please indicate what were the typical intervals between the visits and what was the time interval between T1 and T2. In the methods section, it would be useful to indicate the duration of the visits.
- time interval and duration of the visits were added in the new manuscript: “Patients were visited at T0, T1 (after 1 week) and two more visits were performed after one month from T1 and after three months from T1. Moreover, after 6 months from T1 a final visit was performed (T2). Clinical parameters were evaluated only at T1 and T2. Each dental visit had an average time of 30 to 45 minutes”.
Line 134:Please explain the abbreviation DMFT.
- DMFT abbreviation was explained.
Lines 171 -186: You present the same results two times, just in slightly different words. Firstly, in lines 170 – 177. Then, the lines 178-186 speak about the same. Please decide for one version of result presentation and delete the second one. Since in the table you present exact p-values, I strongly suggest to present the same values also in the text to avoid confusion of the readers. This applies for both significant and non-significant values.
- lines 170-177 were removed in order to avoid repetition in the results. Moreover, exact p values were added in the text.
Line 187: In the table, for the Brushing frequency add the fourth decimal in p-value, so that the numbers are presented in a uniform way. As the means are presented as numbers with 2 decimal places, I suggest to present also the standard deviations in similar way, rounded to 2 decimal places.
- fourth decimals were added of p values and 2 decimals were added for mean values.
Line 192: The reference 12 is inserted in inappropriate place, as you speak about your sample. If you wish to refer to literature, please insert additional sentence citing the reference. Also, the paper you refer to was published in 1998, and since that time a great progress in treatments for autism has been done. Please refer to more recent articles. This suggestion applies also to reference 14, 16, 19. Literature on the topics is available.
- in the new manuscript, we refer to more recent articles and references 12, 14, 16, 19 were deleted.
Line 193: It is recommended to prefer the term „the child with ASD“, instead of „autistic child”.
- child with ASD was added
Line 200: Please rephrase the first sentence. It is the difference that resulted nin-significant.
- the sentence was modified into” The difference in frequency of sugary snacks drinks intake between T1 and T2 resulted non-significant”.
Lines 200-206: The non-significant change in intake of the snacks can be explained mainly by the food selectivity and challenging food behaviors of children with ASD. They have often preferences for unhealthy foods and they stick to their routines. It is very difficult to change their food habits. Interestingly, ABA is the method effective in changing food habits. In ABA, extremely small quantities of snacks are used, in size of millimeters. You may point that it might be useful to for ABA other than sweet reinforcers. Ver, as ABA is one of few evidence-based interventions for ASD, it is not correct to mention exclusively ABA as the only “culprit” that enhances snack consumption in ASD.
- it was written that: “The difference in frequency of sugary snacks drinks intake between T1 and T2 resulted non-significant and it can be explained mainly by the food selectivity and challenging food behaviors of children with ASD; these children have often preferences for unhealthy foods and they stick to their routines therefore it is very difficult to change their food habits. Moreover, .... even if extremely small quantities of snacks are used, in size of millimeters. This practice mostly happens in the initial phase of Applied Behavior Analysis (ABA) therapy and it might be useful to adopt other reinforcers than sweet reinforcers”.
Line 213: the Reference 19 is from 1985. There is more up-to-date knowledge on the topic.
- a more recent reference was added in the new manuscript.
Round 2
Reviewer 2 Report
Dear authors,
unfortunately in my opinion, the work has substantial defects in the methodology, in the standardization and correct execution of the research, the reviews performed are approximate and contain many errors.
Author Response
Dear reviewer,
we have tried to modify the new manuscript according to your comments of round 1 and in the new manuscript we have improved the reviews performed after round 1.
We have carefully considered and made every effort to address our manuscript to your indication.
Kind regards
Reviewer 3 Report
Dear Authors,
I appreciate that you accepted the previous comments. There is still a need to reconsider some formulations in the modified text, please see the comments in the attachment. Although there are many comments, in my opinion the article presents a valuable experience for future readers.
Best regards
Reviewer
